# Association Between Differential Heterogeneity of Antibiotics Consumption and Share of Resistant Pathogens and Its Implication for Antibiotic Stewardship in a German Hospital Intensive Care Unit

**DOI:** 10.3390/antibiotics14121266

**Published:** 2025-12-15

**Authors:** Hans H. Diebner, Pierre Schumacher, Tim Rahmel, Michael Adamzik, Nina Timmesfeld, Hartmuth Nowak

**Affiliations:** 1Department for Medical Informatics, Biometry and Epidemiology, Ruhr-University Bochum, 44801 Bochum, Germany; pierre.schumacher@ruhr-uni-bochum.de (P.S.); nina.timmesfeld@ruhr-uni-bochum.de (N.T.); 2Department of Anesthesiology, Intensive Care Medicine and Pain Therapy, Ruhr-University Bochum, Knappschaft Kliniken University Hospital Bochum, 44892 Bochum, Germany; tim.rahmel@kk-bochum.de (T.R.);; 3Department of Anesthesiology, Intensive Care Medicine and Pain Therapy, Center for Artificial Intelligence, Medical Informatics and Data Science, Ruhr-University Bochum, Knappschaft Kliniken University Hospital Bochum, 44892 Bochum, Germany; hartmuth.nowak@knappschaft-kliniken.de

**Keywords:** antimicrobial stewardship, antimicrobial resistance, antimicrobial heterogeneity, Shannon entropy, differential entropy, Gini index, mathematical modeling

## Abstract

**Background:** The rapid rise in antimicrobial resistance has become one of the 10 most pressing health problems worldwide in recent years. Antibiotic stewardship offers hope in the fight against antibiotic resistance, but it is currently still falling short of expectations. A better understanding of the dynamics of the interaction between antibiotic consumption and the emergence and spread of resistance is urgently needed. **Methods:** We discuss a simple dynamic model based on a differential equation to describe the increase in the proportion of a pathogen’s antimicrobial resistance to an antibiotic as a function of the time-dependent consumption of that antibiotic. Furthermore, we investigate the association of heterogeneity in the consumption of antibiotics with the rate of resistant pathogens. Data basis is the hospital information system and the patient data-management system of a German hospital, restricted to the intensive care unit. To quantify heterogeneity, we discuss and compare different entropy measures. **Results:** For some pathogen–antibiotic pairs, the consumption-dependent dynamic model for the growth in the proportion of antimicrobial resistance provides acceptable predictions, while for others, the model is less suitable. Cross-resistance and complex interactions with other pathogens and antibiotics may be responsible for this, suggesting that the observed dynamic behavior should be complementary, described using heterogeneity models. Time courses of Shannon entropy, the Antibiotic Heterogeneity Index, and the negative Gini Index correlate positively with the time series of the resistance rate. Thus, an increase in heterogeneity correlates with a decreasing resistance rate. However, a time-delayed cross-correlation of a differential entropy measure with resistance share suggests a functional dependence that can be utilized for antibiotic stewardship. **Conclusions:** Evidence is provided that the amount of consumption of certain antibiotics drives the corresponding proportions of pathogens’ resistance to these antibiotics; however, the model predictions of these univariable models are generally not sufficiently good, pointing to a more complex interaction dynamics. Therefore, we switch to the level of structural features and show that the degree of constantly mixing of the shares of antibiotic consumption has a control function regarding the incidence of resistance. Controlling differential consumption heterogeneity, therefore, appears to be a feasible operational basis for antibiotic stewardship. Experimental studies are demanded to identify functional dependencies; however, the integration of clinical expertise with model-based prediction appears to be a feasible antibiotic stewardship strategy.

## 1. Introduction

For critically ill patients in the intensive care unit with severe bacterial infections, the rapid and targeted initiation of adequate antibiotic therapy is of utmost importance for the patient’s outcome. However, antibiotic resistance has an increasingly important influence on the effectiveness of therapy and the chances of clinical recovery. The current worldwide increase in antibiotic-resistant bacteria has therefore been defined by the World Health Organization (WHO) as one of the 10 greatest global health threats to humanity [1,2,3], which requires decisive action through differentiated measures.

Antibiotic resistance is mainly caused by overuse or misuse of antibiotics. The choice of substance used plays the major role here, but dosage aspects and the resulting low antibiotic levels in the blood or the target organ system are also relevant, as this can lead to the selection of resistant subpopulations of pathogens under antibiotic therapy. The choice of substance as well as the appropriate dosage of antibiotics is therefore not only important for patient outcome, but also for the development of resistance itself. Therefore, structured optimization measures against antibiotic resistance are of utmost medical importance. In addition to the pharmacological developments, the rational and responsible use of antibiotics is particularly important. This preventive approach to avoiding resistance is subsumed under the term Antibiotic Stewardship (ABS), also called antimicrobial stewardship (e.g., [4]).

The aim of ABS is to treat patients in the best possible way and, at the same time, prevent selection processes and resistance from occurring in the bacteria. To this end, it would be desirable to regularly evaluate and optimize the rational use of antimicrobial substances with an integrative approach that combines routine intensive care and microbiological data from the clinic with mathematical and methodological analyses: a clinical decision support system. To establish such a system, however, the hurdles arising from the complexity of the clinical situation, as briefly outlined below, must be overcome.

At the beginning of an ABS with a chance of success is the clear identification of the infection. In addition to the actual microbiological identification of the pathogen, ABS includes, in particular, the selection of the appropriate antibiotic, including pharmacokinetic and pharmacodynamic aspects. Clinically, the efficacy of a substance is determined by measuring the minimum inhibitory concentration (MIC). The MIC is the lowest effective concentration of an antibiotic that still prevents the replication of a pathogen in a culture. Clinical threshold values are defined for the respective pathogens (in Europe by the EUCAST—“European Committee on Antimicrobial Susceptibility Testing”), then identify a pathogen as resistant (R) or sensitive (S) to the respective antibiotic tested and support clinicians in selecting the correct substance in the form of the so-called antibiogram [5]. However, the aspect of the active substance concentration achieved in the target compartment is also of decisive importance [6,7].

Attempts to ensure sufficiently high antibiotic levels and thus a sufficient dosage have been made, for example, by determining the blood or tissue levels of a substance as part of therapeutic drug monitoring (TDM) and any resulting dose adjustments [8,9]. However, a third category “I” in antibiograms (English for “increased exposure”), also addresses the requirement for sufficient antibiotic levels, which clinically means that the substance has been tested as sensitive but requires an increased dosage. Regarding the overall cohort, the local resistance situation in the form of ward and/or hospital-specific resistance statistics also plays a relevant role.

As mentioned, the clinical requirements of ABS also include careful and appropriate microbiological diagnostics. A comprehensive prevalence and mortality study recently addressed which pathogens are the most problematic cases [3]. In summary, in 2019, approximately 14% of all deaths were due to a bacterial infection, based on the 33 most important pathogens. Furthermore, 56% of sepsis-associated deaths died as a result of these 33 infections. It is also noteworthy that 55% of deaths from the 33 bacterial species were due to infections of

*Staphylococcus aureus*,*Escherichia coli*,*Streptococcus pneumoniae*,*Klebsiella pneumoniae*,*Pseudomonas aeruginosa*.

Previously, the six pathogens from the so-called ESKAPE series
*Enterococcus faecium**Staphylococcus aureus**Klebsiella pneumoniae**Acinetobacter baumannii**Pseudomonas aeruginosa**Enterobacter*
were discussed as particularly critical with the highest clinical relevance [2]. It almost goes without saying that successful ABS must keep an eye on the prevalence of these pathogens, which are classified as particularly dangerous, especially with regard to the development of resistance.

In a recent article, we presented the first steps toward harnessing the complex dynamics of antibiotic resistance [10]. These analyses have been based on the clinical and microbiological data of a German hospital over an observation period of more than 7 years, which we evaluated descriptively and semi-quantitatively in order to obtain a basis for informed and intelligent action in terms of antibiotic stewardship. The main focus was on the particularly dangerous pathogens mentioned above. The aim of the present work is to extend the results from the recent study and deepen the insights by analyzing the same data set with a focus on the heterogeneity of antibiotic use.

So far, we have observed an increase in the resistance rate with increasing overall consumption, while increases over time, independent of consumption, are fairly moderate. Vancomycin and cefuroxim turned out to be exceptions in the development of resistance, as resistance to these substances appears to decrease with increasing consumption. However, there have been substantial dose adjustments for these substances, which are likely to be decisive here. An intra-host increase in resistance due to treatment time on the one hand and repeated treatments on the other has been observed.

Within the sub-cohort of ineffectively treated patients due to resistance, mortality increased on average, but with ampicillin/sulbactam being a striking exception. Patients with infections caused by ampicillin-resistant bacteria turned out to have a lower mortality rate. Globally, 5 million infection-related deaths per year are attributed to antimicrobial resistance (cf. [11]). However, this does not rule out the possibility of sporadic resistant germ variants that are less deadly than the susceptible variants. Interaction with a presumably greatly reduced microbiome is more likely. Further confounders can only be speculated upon.

The observed resistance rates of the eight most frequently administered antibiotics showed a temporal variability that includes random fluctuations as well as decidedly regular cycles. It has been argued that, in terms of evolutionary dynamics, it seems plausible that the proportion of resistant pathogen variants in a population follows a logistic growth process toward a carrying capacity, i.e., a stable equilibrium [11]. The fluctuations we observed do not support this presumably oversimplified evolutionary dynamic, which can at best be maintained with constant or even increasing consumption of the corresponding antibiotic. Rather, it seems likely that with declining antibiotic consumption, the susceptible pathogen variant can gain an evolutionary advantage. In other words, the development of resistance is not a strictly irreversible evolutionary process. The concept of equilibrium in the sense of an asymptotically stable fixed point of one-dimensional logistic dynamics does not reflect the coupled dynamics of antibiotic consumption and evolutionary resistance development. The assumption tested by the authors [11] that carrying capacity correlates with consumption, i.e., that environmental capacity is a function of consumption, is not particularly meaningful.

However, we have not yet considered individual pathogen–antibiotic pairs (referred to by [11] as bug–drug pairs) in our evaluation, which we make up for in this publication. Specifically, we present a simple model that couples the “idle” dynamics in the form of a logistic growth process with differential antibiotic consumption. We consider this type of coupling to be more appropriate than a direct dependence of a presumed carrying capacity on consumption. In this regard, it is worth noting that the proportion of resistance to piperacillin/tazobactam across all pathogens shows remarkable constancy over time [10], suggesting that the (total) trajectory in this special case has indeed stabilized at a carrying capacity that has already been reached. This is not a contradiction for the case of a constant consumption over the entire observation time.

Also shown previously (cf. [10]), the time series associated with the various antibiotics showed pairwise time lag correlations, which indicates the existence of retardedly mediated cross-resistance. In particular, we refer to these last-mentioned observations of complex time-delayed interactions in order to take a closer look at these relationships specifically regarding possible resonances between dynamic consumption patterns and the incidence of resistance.

In this context, it is worth recalling the idea of cyclical variation in antibiotic use to suppress resistance without dispensing with antibiotic treatment and without the pressure to constantly develop new active agents. We therefore revisit the most important basic ideas and existing work already described in our recently published papers to outline the background [10,12]. Cyclical allocation of antibiotics or so-called mixing (other terms are in circulation) are strategies at hospitals with the aim of contributing to a reduction in the prevalence of resistant germs by means of spatial (between departments, mixing) or cyclical (temporal) variations in the proportions of consumption of different antibiotic groups. According to a systematic review [13], there is only one randomized controlled trial (RCT) on the effectiveness of such strategies [14]. However, there are some studies that at least compared systematic cyclical administration strategies and standard administration between clinics or carried out before-and-after comparisons (cross-over) [13,15]. In the RCT study, cycling performed worse than the control ABS [13,14].

Noteworthy, in numerous studies, “mixing” was chosen as the control strategy, i.e., an alternating exchange between departments. Cycling, i.e., a temporal periodic change simultaneously across all departments, showed no difference to the mixing procedure, which is not surprising from a theoretical point of view if one assumes sufficiently isolated conditions between the departments. Nevertheless, all studies (RCT and cohort studies), including the cross-over studies, were considered together in the meta-analyses, regardless of whether they tested against “mixing” or “without strategy”, which must be viewed very critically and neglects the important separate consideration of mixing and cycling. Of note, in the meta-analysis by [15], a distinction was made between the two control strategies as part of a secondary analysis. In Gram-positive bacteria, the cycling strategy showed a slightly stronger effect in terms of avoiding resistance. It appears that there is no general evaluation independent of other biological and medical boundary conditions. It is, therefore, possible that the cycling concept itself is not well thought out. The question of whether cycling or mixing contributes to the “rational and responsible use of antibiotics” has therefore not been conclusively clarified.

Cycling according to a fixed scheme (scheduled cycling) means that the informed, i.e., rational use of antibiotics is deliberately avoided, so that conceptually a control rather than an intervention strategy is defined here. In this context, it is worth recalling that in a seminal cycling study published by Gerding et al. [16], the authors investigated interrupted time series in the administration of aminoclycosides. These irregular antibiotic prescriptions followed an observed pattern in the emergence of resistant germs rather than a fixed periodic cycling scheme. In retrospect, it appears that the authors intuitively used the method that is now referred to as “clinical cycling” and which is actually superior to scheduled cycling because it is based on clinical evidence for the necessity of adapting the antibiotic consumption. Unsurprisingly, scheduled cycling is explicitly not recommended in a German S3 guideline [17], whose validity has expired 31 January 2024, by the way.

Despite the aforementioned counterarguments, the poor performance of “scheduled cycling” does not speak against cycling *per se*, but rather against cycling that is not carried out intelligently. One could even say that conceptually, scheduled cycling corresponds more to a “placebo-like” (control) ABS, whereas clinical cycling based on clinical expertise corresponds to the intervention group. In the context of this interpretation, the many studies on cycling are in fact rather evaluation procedures for assessing whether the respective “clinical cycling” is viable based on implicit clinical knowledge in comparison to a non-informed cycling strategy. In other words, despite the clinically contraindicated settings of scheduled cycling programs, these strategies represent a kind of basic model structure whose quantitative explanation is the basis for the description of more complex switching strategies (clinical cycling).

In our own preliminary work [12], we have created and published a mathematical framework that is suitable for adequately quantifying the effect of clinical cycling and, in borderline cases, scheduled cycling. Subsequent correlation analyses revealed a relationship between the heterogeneity of antibiotic consumption and the prevalence of resistant pathogens, indicating a reduction in the prevalence of resistant germs. The heterogeneity changes on the pathogen side follow the changes on the consumption side. It is worth mentioning that in most of the older studies conducted, a quantification of the degree of mixing or cyclic variation has rarely been used. An exception is the study by Sandiumenge et al. [18], in which an Antibiotic Heterogeneity Index (AHI) was used. This work was followed by a few publications in which AHI or a similar quantification of heterogeneity was discussed (e.g., [19]), but if so, then only in passing. Of note, AHI is invariant to swapping the antibiotic classes. This means that if the consumption shares of two antibiotics are swapped, AHI remains unchanged. This must be kept in mind when discussing cycling strategies based on AHI. The view expressed in publication [20] that the focus in ABS on strategies of cyclical prescribing has shifted to a focus on heterogeneity sounds as if these were disjoint strategies. However, when correctly quantified, variations in the administration of antibiotics, including cyclical prescribing, result in a corresponding change in heterogeneity.

Measures of heterogeneity or, more generally, of diversity are frequently used in ecological studies to calculate “mixing”, i.e., the degree of heterogeneity. These measures are related to entropies, which originated in statistical physics for the quantitative description of mixing processes (dispersions) and similar phenomena. The Shannon entropy known from information theory is, like AHI, a global entropy, i.e., invariant to permutations of the species. Local heterogeneity measures should be used in order to be able to record changes over time. The Kullback–Leibler entropy is such a local entropy and has proven its worth in our preliminary work in the context of ABS as well as of spatio-temporal epidemic patterns [12,21]. However, the concrete choice of the final form of the analysis algorithms depends on specific conditions.

Theoretically, it would be conceivable to extrapolate the observed mixing states of antibiotic consumption and predict the resistance incidence. Alternatively, it appears to be appealing to create possible interaction scenarios, in such a way that an “optimal cycling regime” defined by the best possible reduction of resistance is achieved. However, such a project is a real challenge due to the necessary constraints, such as those imposed by biological and clinical conditions and regulations. Our preliminary work represents an important step in this direction, also taking into account clinical constraints (antibiogram, pharmacokinetics, and microbiological parameters, guidelines) [8,9,10,12,21]. In this publication, an exploratory observational study, we explore the relationship between antibiotic consumption heterogeneity and resistance emergence for the intensive care unit of a German University hospital, located in Bochum.

## 2. Results

### 2.1. Basic Summary Statistics

Complementary to our recently published evaluation [10], in this paper, we focus on the evolutionary dynamics of resistance development and its dependence on antibiotic consumption. Despite this new focus, we consider it useful for better readability to again explain the basic structure of the available data in somewhat greater detail than is strictly necessary. As described in detail in Section 4, the data body of the primary clinical systems database is fragmented. A descriptive overview of the relevant data fragments is provided below in the form of summary or frequency tables, respectively.

#### 2.1.1. Demographic Data

In addition to the patient’s sex and age, demographic data also includes the length of stay in the hospital and ICU, as well as the discharge destination and discharge diagnosis. In total, n=7718 patients are contained in the dataset; however, some patients had multiple hospital stays. The instances (subjects) considered in the following analysis are therefore disjunct intervals of stay, not the patients themselves.

#### 2.1.2. Documented Antibiotic Administration

The second dataset contains, for each antibiotic treatment day per patient, the antibiotic consumption. In total, n=67 different forms of administration of antibiotics have been in use. Most antibiotics have been applied intravenously (i.v.); however, occasionally other forms of application, such as orally (p.o.), by enema, or by inhalation, have alternatively been in use. This leaves n=57 unique antimicrobials in use.

The nine most frequently administered types of antimicrobial treatments are listed in Table 1. The second column contains the cumulative days of use, while the third column contains the number of individuals who received the corresponding antibiotic. In our analysis, we focus on these frequently administered antibiotics, except for flucloxacillin (since no antibiograms exist for this drug) and caspofungin (antimycotic), leaving seven evaluable antibiotics.

In addition to the total consumption of these seven most frequently used antibiotics over the entire observation period, Appendix A shows the time series of monthly aggregated consumption of these antibiotics. The consumption of ciprofloxacin is gradually declining towards the end of the observation period, which can be explained by the safety concerns raised in a so-called red-hand letter from the competent authorities [22]. Moreover, during the second half of the observation period, meropenem (a last resort carbapeneme antibiotic) apparently rapidly catches up with piperacillin (a broad-spectrum β-lactam antibiotic) in terms of importance.

#### 2.1.3. Demographic Characteristics Summarized by Antibiotic Treatment Indicator

After merging the demographic data with the dataset containing the antibiotic treatments, it is possible to determine for each distinct sequence of stays per patient (ward stays) whether or not an antibiotic was administered during these ward stays and generate a corresponding binary indicator. Summary Table 2 shows the demographic characteristics stratified by this AB-treatment indicator (“w/o AB” vs. “with AB”). According to the division of the individual total stay times into clearly separated stay intervals, the subject entity of Table 2 is ward stay (n=9201) rather than an individual person (n=7718).

To briefly summarize our previous findings here once again, male patients are slightly more likely to be treated with antibiotics. Age is not a determinant of being treated or not. As was to be expected due to the additional vulnerability caused by nosocomial infections, death is a more likely destination among the antimicrobially treated patients, whereby AB-treatment is conceived as a surrogate for infection. In addition, the duration of hospitalization as well as the length of stay at the ICU are significantly longer for the AB-treated cohort. For the results of extensive regression analyses, please refer to [10].

#### 2.1.4. Antibiogram

The antibiogram data contain 155,594 exposures of isolated pathogens to antibiotics. The majority yielded susceptibility, and the very few antibiotic challenges did not lead to any usable results. See frequency Table 3 for a complete summary, showing that approximately 27% of the exposures fall into the resistance class. The 10 most frequently isolated pathogens are shown in Table 4.

Skipping the two opportunistic pathogenic yeast species, *Candida albicans* and *Candida glabrata*, we identify the most dangerous and problematic bacteria from the ESKAPE series and those associated with high mortality rate ([2,3]) and, in addition, *Staphylococcus epidermidis*. Thus, in terms of pathogen prevalence, the clinic studied shows the known global characteristics.

In addition to the overall frequency of isolated pathogens (Table 4), the supplement also shows the time series of monthly aggregated detected cases for the eight most common pathogens (see Appendix A). Striking declines coinciding with the onset of the COVID-19 pandemic in the number of observed cases of *Staphylococcus epidermidis*, *Enterococcus faecalis*, and *Staphylococcus aureus* can be observed. The protective measures during the COVID-19 pandemic could be an explanation, albeit speculative [23,24]. Worth of note, the authors of [24] refer to the reduced incidence of infections with carbapenem-resistant and methicillin-resistant pathogens during the first two pandemic years, but without referring to the denominator of the total infection, which is why this prevalence study is of little use.

### 2.2. Shares of Antibiotic Consumption over Time

The shares of monthly aggregated antibiotic consumption, $p_{A}\left(t\right)$ (cf. Equation (Equation 5)), are depicted in the form of a stacked barplot over time for the entire set of administered antibiotics A in Figure 1, and for a selection of the seven most frequently applied antibiotics (cf. Table 1) in Figure 2. Of note, flucloxacillin has been skipped from the shortlist shown in Table 1 since no antibiograms were created for this drug. Furthermore, caspofungin is an antimycotic and is also excluded from the selection. Resistance does not appear to be a pressing problem for caspofungin, anyway. Different methods of administration of the same antibiotic are summarized here, but only after calculating the consumptions for each method of administration normalized to the respective medians.

Based on the color codes assigned to the various substances, it is immediately apparent that the respective shares of consumption fluctuate considerably over time. In particular, Figure 2 shows that the proportion of consumption of ciprofloxacin is gradually fading out, likely due to the red-hand letter indicating unacceptable adverse events [22]. Likewise, the proportion of consumption of linezolid decreases markedly over time. These decays are compensated for by an increase in the proportions of consumption of the remaining substances. In addition to these particularly noticeable fluctuations, however, the other shares also vary considerably.

### 2.3. Modeling the Evolution of Resistance Rates

This section serves to describe the evolution of the resistance rates of the most common pathogens with respect to the most commonly used antibiotics by means of a deterministic dynamical model. Inspired by a recent publication [11], we test the assumption that the evolution of the resistance rate per pathogen–antibiotic pair follows a logistic saturation dynamic.

To start with, for each pathogen–antibiotic pair from the group of the eight most common pathogens (cf. Section 2.1.4) and the group of the seven most commonly administered antibiotics (cf. Section 2.1.2) for which antibiograms had been created, the observed time series of the respective resistance rates calculated per month are shown in Figure 3. Obviously, antibiograms have not been created for all 8×7 possible pairs. For example, *Staphylococcus aureus* has only been exposed to vancomycin and did not show any resistance over the whole observation time. On the contrary, other pairs as *E. cloacae*-ampicillin/sulbactam yield 100% resistance throughout the observation time. For some pathogen–antibiotic pairs, antibiograms have only been created over very short periods, leaving them unusable for our purposes.

The antibiograms of the pathogens *Pseudomonas aeruginosa* (exposed to 3 antibiotics), *Staphylococcus epidermidis* (exposed to 1 usable antibiotic), *Escherichia coli* (exposed to 5 antibiotics), and *Klebsiella pneumoniae* (exposed to 5 antibiotics) are suitable for adapting a dynamic model to the curves. However, the observed aperiodic oscillating or strongly fluctuating time courses already show that simple logistic growth dynamics (see Equation (Equation 3)) are not an adequate option.

However, if the model is extended with a driver dynamic for the resistance-modulating impact of antibiotic consumption (cf. Equation (Equation 4)), some of the observed time series can be reproduced with astonishing accuracy, as depicted in Figure 4 for the case of the three *P. aeruginosa*–antibiotic pairs. Regarding meropenem exposure in particular, the highly fluctuating resistance rate can be modeled surprisingly well by the growth-controlling effect of differential meropenem consumption.

With regard to ciprofloxacin, it should be noted that consumption was drastically reduced in the second half of the observation period (described in [10], also cf. Appendix A) due to safety concerns raised in a red-hand letter [22], which prevents a good model fit, but the overall trend is still well reflected. The same holds for the rate of resistance to piperacillin/tazobactam. The predicted wave troughs and wave crests follow the observed ones, albeit with a significantly smaller amplitude.

The temporal curve of resistance of *Staphylococcus epidermidis* to linezolid exhibits strongly pronounced apparently irregular “bursts”, which are not adequately represented by the consumption-controlled logistic growth model, as depicted in Figure 5. These “bursts” are likely the result of significantly more complex interacting mechanisms, which can at best be speculated about. Since no information is available regarding laboratory guidelines or compliance with them, detection bias cannot be ruled out, in addition to biochemical complications such as cross-resistance.

Regarding the total of four bug–drug pairs of *P. aeruginosa*–antibiotics and *S. epidermidis*–linezolid, estimated model parameters are depicted in Table 5. The logarithmic parameters of the consumption-triggered logistic growth model (column 4) were estimated using a maximum likelihood method under the assumption of a log-normal distribution. The large standard errors (column 5), which could not even be determined in one estimation process, show that the likelihood function is rather flat. Most likely, there are numerous local maxima, which prevent the determination of confidence intervals by likelihood profiling. A look at the table with the parameter values and the *p*-values shows once again that the two pairs *P. aeruginosa*–ciprofloxacin and *P. aeruginosa*–meropenem lead to acceptable fits and, in particular, that the consumption dependency becomes significant. In other words, the effect of control through consumption, expressed by the parameter α, proves to be particularly striking here.

The fact that the evolution of resistance rates displays pronounced waves rather than following a simple logistic saturation dynamic is also confirmed by the pathogen–antibiotic pairs of *Escherichia coli* and *Klebsiella pneumoniae*, as shown in Figure 6. However, some curves from the set of 10 time series (5 per pathogen) are predicted quite accurately by the extended model, i.e., the model triggered by antibiotic consumption. Some other curves are not well reproduced by these “univariable” models.

The estimated parameters of the models describing the resistance curves belonging to *Escherichia coli* and *Klebsiella pneumoniae* are listed in Table 6. For these two pathogens, the purely linear (univariable) effects of the corresponding consumption are very low in terms of resistance to all antibiotics tested. The corresponding estimated values for α are tiny, with some extremely large standard errors. In the case of cefuroxim, the maximum likelihood routine does not converge sufficiently well (NAs are generated). The limitations of the simplified dynamic model are clear here.

Given the plausible assumption that the dynamics of resistance development are more likely to be the result of a synergistic interaction between all pathogen–antibiotic pairs, the occasionally observed linear functional relationships are rather surprising. Cross-resistance, to name a phenomenon that is now much discussed [25,26,27], would require modeling using multiple coupled differential equations. However, the available antibiogram data from only one clinic is far too sparse to perform a corresponding model validation.

Bergstrom et al. [28] attempted to develop a more complex dynamic model to describe the evolution of resistance. However, this model included only a few variables, both in terms of antibiotics and pathogens. Adapting the model to data from cycling studies did not provide any evidence of a reduction in the emergence of resistance. However, the results from applying the model to mixing studies suggest a moderate reduction in mixing strategies. However, the negative results of evaluations of scheduled cycling studies do not in any way rule out the effectiveness of informed rational clinical cycling as addressed here. To conclude, the adjustment and verification of a more complex model than that of Bergstrom et al. [28] does not seem feasible given the limited amount of data available.

In this context, one of the results in [10] deserves attention. If the resistance rates for a particular antibiotic are aggregated across all pathogens that exhibit this resistance, a clear dependence on consumption can be demonstrated using a linear regression model. The cumulative effect thus shows a dependence on consumption, although the dependence appears to be quite heterogeneous with regard to individual pathogens. This suggests modeling that explicitly captures this heterogeneity.

### 2.4. Heterogeneity in Antibiotic Consumption

A reductionist model approach, in which the dynamics of the time courses of resistance proportions associated with individual pathogen–antibiotic pairs are modeled only linearly, i.e., without mutual coupling, falls short, as we have seen. At the same time, systemic, more complex modeling based on differential equations does not seem feasible without significantly more extensive data for empirical validation. Therefore, in order to grasp the complexity of the interactions between antibiotic use and the evolution of resistance, we switch to a higher phenomenological level of description, so to speak. Specifically, we capture heterogeneity using appropriate mathematical tools, namely entropy measures, and use them to model the control of resistance development.

Estimates of different measures of heterogeneity as introduced in the Methods Section 4.3.5 are depicted in Figure 7. The corresponding time series of entropies applied to the entire set of relevant antibiotics (cf. Figure 1) are shown in the left column of Figure 7, whereas the panels shown on the right-hand side refer to the selected set of the seven most frequently administered substances (cf. Figure 2). The entropies were calculated using monthly aggregated antibiotic consumption and then smoothed over a 24-month window using a moving average algorithm.

The Shannon entropy, $H_{S}$, initially starts at a relatively high value of approximately 0.75 for the full set of antibiotics and 0.93 for the short list, respectively, i.e., not too far from the uniform distribution of consumption shares. Due to the non-linear scaling of the Shannon entropy, the initial value of AHI, which is approximately 0.42 for the full set of antibiotics and roughly 0.76 for the restricted set of 7 antibiotics, respectively, differs from the Shannon value due to deviating scaling. Taking into account that the Gini Index, GI, scales inversely, its time course shows a similar shape to Shannon entropy and AHI.

Strictly speaking, Shannon entropy is not defined when vanishing probabilities/ proportions occur due to the logarithmic expression on which the definition is based. On the other hand, the weightings of individual very large or very small proportions are particularly important in some applications in terms of their contribution to entropy. If the three heterogeneity measures are made comparable by means of a z-transformation, it becomes apparent that the three measures diverge particularly at very small entropy values. This fact is illustrated in Appendix A.

It follows that for the three heterogeneity measures, Shannon entropy, AHI, and Gini Index, the shapes of the curves are similar, and the degree of homogeneity drops moderately for approx. 40–50 months, followed by a rather steep drop of approx. 0.1 units within a few months until month 70. With respect to the full set of antibiotics, a rather abrupt rise from month 75 on until the end of the observation period can be observed, obviously resulting from changes in consumption of the less frequently administered antibiotics. One possible explanation of the observed behavior is that some substance classes, such as the fluoroquinolones, were almost completely removed from the repertoire in the course of time; thus, there was a concentration in consumption on the remaining antibiotics, roughly during the second half of the observation period. There, therefore, appear to be two phases in antibiotic consumption. As will be shown below, this jump in the temporal consumption pattern is actually reflected in the emergence of resistance.

However, if the distribution in the first month is used as a reference, an alternating deviation of the distributions from this reference can be observed, reflected by an oscillating entropy time course. This can be seen in the fourth row of Figure 7, which shows the course of the Kullback–Leibler divergence related to the initial distribution, $KL_{t_{0}}$. A similar oscillating pattern can also be seen for the entropy measures AHIΔ and KLΔ≡$H_{S}\Delta$, which relate to the distributions of the respective previous months. This behavior suggests that there are constant changes in the respective consumption shares of the individual antibiotics, even if the overall heterogeneity changes only slightly. Such a differential method of determining heterogeneity seems predestined to capture temporal mixing behavior and could therefore be the preferred method for evaluation in the context of ABS compared to a measure of static heterogeneity. In the following sections, the temporal behavior of the heterogeneity in antibiotic consumption is related to the evolution of antimicrobial resistance.

### 2.5. Correlations Between Heterogeneity of Antibiotic Consumption and Proportion of Resistance

The time course of the proportion of all antibiograms (Equation (Equation 2)) that have been produced for all pathogen–antibiotic pairs out of the set of the eight most frequently isolated pathogens and the seven most frequently administered antibiotics (cf. Table 1, without antifungals, and Table 4), which have been detected as resistant, is depicted in Figure 8. An oscillating pattern between approximately 19% and 26% antibiograms labeled resistant can be observed, however, with a declining overall trend until month 75.

#### 2.5.1. Differential but Not the Static Consumption Shannon Entropy Leads the Share of Resistance

Pearson’s correlation coefficient is estimated as ρ=0.902 (p<0.001) for the cross-correlation between the share of resistance (cf. Figure 8) and AHI, and ρ=0.866 (p<0.001) for the Shannon entropy, ρ=0.886 (p<0.001) for the negative Gini Index, respectively (see Figure 9 for the entropies). For all three cases, a positively time lagged $p_{R}\left(t+dt\right),dt>0$ leads to a decrease in correlation with the entropy at time *t* (cf. Figure 9). A proper interpretation is far from straightforward, although this behavior suggests that a current high level of resistance leads to a higher heterogeneity in consumption in the near future. In other words, heterogeneity lags behind the evolution of resistance, whereas the opposite would actually be desirable. It can be concluded with due caution that static entropy as a measure of heterogeneity is not a suitable predictive measure for the development of resistance.

On the contrary, the cross-correlation of differential entropy measures $H_{S}\Delta$ and −AHIΔ, respectively, with the proportion of resistance increase considerably for increasing time lags dt of $p_{R}\left(t+dt\right)$ within roughly 6–7 months (see Figure 9). Although this does not in any way prove a mechanistic effect, both the strong correlation and the delay do not appear implausible and suggest a functional relationship. In other words, it appears to be feasible to control the evolution of resistance at least to some extent by varying the consumption shares of antibiotics. Apparently, it takes some months until the changes in consumption have an impact on the generation or suppression of resistance. Once again, great caution is required in interpreting these data, but it appears that it is not the high level of heterogeneity currently observed that is the decisive control measure, but rather the ongoing change in the proportions of antibiotics consumed. This suggests an ongoing permutation of consumption patterns, which, however, does not necessarily have to be strictly cyclical. The effect of these changes appears to be delayed by around 5–6 months, which also indicates that the scheduled quarterly cycling used in earlier studies is inappropriate and that clinical cycling at a somewhat slower pace should be prioritized. Finally, differential Shannon entropy captures changes in antibiotic administration better than differential AHI and is therefore the more sensitive measure for this purpose.

#### 2.5.2. Phase Space Construction Reveals a Bi-Phasic Time Course

Oscillations with a moderately decreasing mean trend are observed for the time courses for the consumption entropy (in the form of the AHI, Shannon entropy, or Gini Index) on the one hand and for the proportion of resistance on the other. The entropy and consumption time courses correlate, and to a much greater extent, after the entropy curves have been shifted by four months. This is what we conclude thus far from the previous section.

We now show that there is a further, particularly striking non-linear correlation structure. The previously observed jump in the amplitude of the entropy at about 50 months is also found in the resistance curve, so that the trajectory in the reconstructed phase space, spanned by entropy and resistance, has a bi-phasic structure as shown in Figure 10. First, the trajectory oscillates in a restricted area with high entropy and resistance values (dark blue part of the trajectory), and then changes to a restricted area with small amplitudes of both variables (bright red part of the trajectory). This result holds for all three calculated entropy measures and does, therefore, not strongly depend on the concretely chosen measure.

For sensitivity analysis, the proportion of pathogen isolates detected as resistant was recalculated on the assumption that the intermediate (increased exposure) category, I, is equal to the resistant category, R. From a clinical perspective, I belongs to the susceptible class at the maximum permissible dose; however, a change in the definition of I occurred during the observation period in 2019, so it makes sense to perform the proposed sensitivity analysis. The corresponding temporal progression of $p_{R}$ and the associated phase space reconstructions are shown in Appendix A, analogous to Figure 8 and Figure 10. We refer to the explanation in the Appendix A and briefly summarize here that $p_{R}\left(t\right)$ exhibits significantly less pronounced intermittent peaks and that the bi-phasic trajectory is even more strongly separated into two sub-areas in phase space. In summary, the analysis results depend only slightly on whether class I is interpreted as resistant or susceptible.

Moreover, the bi-phasic behavior can also be explored and clearly illustrated by means of a fragmented regression. To this end, a changepoint of the time variable is estimated such that the two regressions restricted to the interval before and the interval after this changepoint, respectively, maximize the likelihood. The convincing result of such a fragmented regression for both the Antibiotic Heterogeneity Index (AHI) and the share of resistance, $p_{R}$, is shown in Appendix A.

## 3. Discussion

We analyzed secondary data from an intensive care unit with regard to the association between antibiotic use and the development of pathogen resistance to these antibiotics. This immediately reveals the greatest limitation, namely that secondary data analyses generally make it difficult to interpret the observed associations in a causal manner. However, we looked at the evolution of antibiotic resistance from a mathematical-dynamic modeling perspective, so that functional dependencies can be inferred with due caution, i.e., only within the scope of the validity of the dynamic models.

In general, it is valid that secondary data analyses using mechanistic models in biology and medicine–meaning models that incorporate well-defined laws–can achieve a similarly high level of evidence as experimental studies. In fact, the field of antibiotic stewardship must be structurally classified as healthcare research or epidemiological research rather than as a field of traditional clinical studies. Controlled and randomized studies are only possible, if at all, as cluster-randomized studies or with sequential intervention and control designs (stepped wedge, cf. [29]). As in healthcare research, there is often an extremely long delay before the intervention takes effect, even if it is significant. This so-called EbM-lag (see e.g., [30]) necessitates the use of pragmatic study designs, preferably incorporating mechanistic models to ensure a high degree of predictability.

In the present study, too, it would be desirable to refer to biochemical laws, but realistically, we are dependent here on the recording and modeling of phenotypic characteristics or proxy variables. The sensitivity levels S, I, and R would be a good example. Using the exact MICs instead of the three-part classification would certainly be an option, but it would require increased effort in the laboratory. We are keeping this option open for upcoming analyses, in particular to better reflect the problematic classification of the middle category “increased exposure.” Relying on biochemical laws at this point would inflate the degree of complexity to an unmanageable level. Nevertheless, we are convinced that even modeling using intrinsic variables that capture a structural state macroscopically can contribute to easily interpretable findings and allow conclusions to be drawn.

Specifically, back to what we did, mutually independent dynamic models designed to predict the evolution of pathogen resistance to antibiotics for each pathogen–antibiotic pair have generally proven to be insufficient. The dynamics of resistance can only be explained reasonably satisfactorily for a few pairs based on these univariable models, whereby in these cases, it is most often the effects of the control parameters describing the transfer from consumption to resistance that become significant. We postulate that there are interactions between pathogen–antibiotic pairs, for example, in the form of cross-resistance. However, empirical validation of suitable coupled dynamic models is not feasible based on the data set, which is too small for this purpose. However, we note that the model-based prediction is particularly good for the combination of *P. aeruginosa* and meropenem. If further studies show that the meropenem resistance of *P. aeruginosa* is independent of the consumption of all other antibiotics, the only stewardship strategy ultimately available is the long-term reduction of meropenem consumption.

Alternatively, we move to a higher phenomenological level, so to speak, and switch to a scalar quantification of the heterogeneity of antibiotic consumption, i.e., to the description of an intrinsic structural feature. We also summarize the resistance of all pathogens. In this way, we can provide some evidence that the evolution of resistance can be favorably influenced, i.e., resistance reduced, by a constant mixing of antibiotic consumption shares. A suitable quantitative representation of the mixing process is provided by the Kullback–Leibler divergence for a momentary distribution of consumption shares, whereby the reference distribution is defined by the distribution in the immediately preceding time interval. This representation is equivalent to Shannon entropy with this very reference instead of a uniform distribution.

The mere statement that constant mixing in antibiotic use is beneficial in terms of resistance development may seem sobering. The scalar quantification of the mixing process, which is therefore subject to massive information loss, is difficult to operationalize. Realistically, however, due to numerous clinical and biological restrictions, only a few antibiotics are suitable for consumption permutations anyway. From our perspective, it makes sense to subject what is known as clinical cycling to much stricter control and to move beyond pure unspoken heuristics into a controlled exploratory phase. It would be desirable for several clinics to coordinate their efforts so that comparisons of different strategies can ultimately be carried out, thereby creating at least a quasi-experimental situation, perhaps even in the form of cluster randomization.

It should be noted that the evaluations of the time series depend to a not entirely negligible extent on the smoothing procedures used. Even monthly aggregation results in a loss of information, but due to the highly fluctuating time series, using smaller base elements, it is no longer possible to evaluate them meaningfully, at least not within the framework of the model approaches used. The smoothing and aggregate levels applied are the result of a compromise between fundamental feasibility and precision. Similarly, the restriction to the most commonly administered antibiotics and the most important pathogens represents a limitation that was unavoidable due to the otherwise unmanageable fluctuations. In the long term, however, antibiotics that have been used less frequently to date should also be included in strategic considerations.

## 4. Material and Methods

### 4.1. Primary Clinical Systems Database

After obtaining an ethics vote on 24 May 2023 for our study “Model-based optimization of antibiotic stewardship strategies (Opti4ABS)” from the Ethics Committee of the Medical Faculty of the Ruhr University Bochum (register number: 23-7836-BR), the clinical data were retrieved. The evaluation included patients who were treated in the interdisciplinary surgical intensive care unit of the University Hospital Knappschaftskrankenhaus Bochum between 1 January 2016 and 31 March 2023. A total of 7718 patients with 9201 ward stays were identified.

A data query was performed from the database of the primary clinical systems (hospital information system and patient data-management system). The raw data included the following entities:Demographic data, including principal diagnosis,secondary diagnoses and procedures,location information (rooms where the patient was in the ICU),documented antibiotic administration,laboratory values/measurements/scores.

These data were then checked for plausibility, corrected if necessary, converted into an initial structured format, and provided in CSV format to the analysts. Before being made available as part of the preparation, the data set was extensively anonymized with the aim of retaining the granularity of the data required for the analysis, but at the same time ensuring that no reference to natural persons could be made at the end. The anonymization, therefore, included the following aspects, among others:Calculation of the age at hospital admission and removal of the date of birth,Removal of the patient-identifying case and patient number,Removal of other personal details (if available in the data record), e.g., surname, first name, address, other identification numbers, etc.,Conversion of the date reference of the individual data records into a treatment day specification from which it is no longer possible to draw conclusions about the date.

For the project, it is necessary to map the spatial and temporal relationship of patients treated on the ward at the same time (e.g., to map the transmission of resistance between patients). In order to anonymize the data set, all dates for each patient were converted into a uniform global treatment day (as a numerical number: 0, 1, 2, … 989, 990, 991, …, etc.). The uniform reference date (day 0) was only known to the clinical study center at the University Hospital Knappschaftskrankenhaus Bochum and was destroyed after completion of the data preparation and analysis process. The date was explicitly not set to 1 January 2016 (start of inclusion of patients), but to a random date thereafter. Similarly, the end date was not 31 March 2023 (end of patient inclusion), but a random date before this. This date was also not communicated externally and was destroyed at the end.

### 4.2. Antibiogram and Microbiological Data

The clinical data set was expanded to include the specific microbiological findings (including resistance determinations). As these data are not primarily held in the hospital information system, it was necessary to program an additional interface to the data-holding system, which automatically extracts the relevant findings and processes them in accordance with standards, as well as anonymizes them. The microbiological data contain information on the pathogens isolated and microbiologically examined, in particular the organ/tissue sampled and the quantity, as well as their susceptibility to antibiotics. In order to create the antibiograms, almost every isolated pathogen of every patient was subjected to multiple antibiotic provocations, and the corresponding sensitivity was determined via the MIC. Only a few pathogens were not challenged.

At the turn of the year from 2018 to 2019, the previously so-called intermediate sensitivity category *I* was redefined by EUCAST as “susceptible, increased exposure” [5]. The definition also changed the application recommendation. Antibiotics classified as “intermediate” were generally not prescribed before 2019, but were prescribed afterwards, albeit at an increased dose. As the observation period of the data examined here begins in 2016, this change in definition must be kept in mind. We address this problem at the appropriate point (see Section 4.3.2) by means of a suitable sensitivity analysis.

### 4.3. Statistical Analysis and Mathematical Modeling

#### 4.3.1. General Notes on Statistics

Descriptive statistics are presented both graphically as well as through frequency or contingency tables, or Pearson correlation (including time lag cross-correlation, cf. [31]) and regression analyses, respectively. The usual 5% significance level is used for comparisons and regressions as well as 95% confidence intervals, when appropriate. The statistical tests used in each case are noted in the footer of the corresponding tables and include chi-square or Fisher’s exact tests for categorical variables, t-tests or, if applicable, rank tests for continuous variables. All analyses and graphs were created using the statistical programming language R version 4.5.1 (13 June 2025) [32].

#### 4.3.2. Proportion of Pathogens Tested as Resistant

Two types of proportions of pathogens tested as resistant are discussed. Thereby, only the eight most frequently tested pathogens and the seven most frequently applied antibiotics (cf. Table 1 and Table 4) are discussed. The following notions are used: P is the set of (the eight most common) pathogens, A is the set of (the most common) antibiotics, n(P,A) is the number of unique challenges of a given P∈P with a given A∈A, and $n_{R}\left(P,A\right)$ is the number of challenges which prove resistance of the tested pathogen P∈P to A∈A. Then,(1)$$p_{R}\left(P,A\right)\left(t\right)=\frac{n_{R}\left(P,A\right)\left(t\right)}{n(P,A)(t)}$$
is the proportion of monthly aggregated challenges that prove resistance of the tested pathogen to the corresponding antibiotic. Time parameter *t* is given in months running from 0 to the maximum length of the observation time in months (86 or 87, respectively, due to missing antibiograms in the last month for some pathogen–antibiotic pairs).

In addition, for an analysis at an aggregated level, the following overall percentage of resistance is also required:(2)$$p_{R}\left(t\right)=\frac{\sum_{P\in P,A\in A}n_{R}\left(P,A\right)\left(t\right)}{\sum_{P\in P,A\in A}n\left(P,A\right)\left(t\right)}.$$
The time series $p_{R}\left(P,A\right)\left(t\right)$ and $p_{R}\left(t\right)$, respectively, as they are ultimately fed into the evaluations performed, are smoothed using a moving average function (central, partially at both ends) with a window of 24 months in order to curb the peaks.

It should be noted once again that the antibiotic sensitivity of pathogens is divided into three levels: R, I, and S. In the main analysis, we combined I and S into one susceptible category for the purpose of dichotomization. This corresponds to the EUCAST definition of I, namely susceptible at increased doses. However, it could be argued that the increased MIC associated with I already indicates resistance. For sensitivity analysis, we combine R and I into one resistance category and calculate $p_{R}\left(t\right)$ based on this classification, whereby the distinction is easily apparent from the respective context.

#### 4.3.3. Evolution of the Proportion of Pathogens Tested as Resistant

Inspired by [11], the assumption that the evolution of the resistance rate per pathogen–antibiotic pair, i.e., $p_{R}\left(P,A\right)\left(t\right)$, follows a logistic saturation dynamic is investigated. Although the logistic growth does have a closed solution, we here use the representation as a differential equation due to its desired expandability:(3)$$\frac{dp_{R}\left(P,A\right)\left(t\right)}{dt}=r\cdot p_{R}\left(P,A\right)\left(t\right)\cdot \left(C-p_{R}\left(P,A\right)\left(t\right)\right),$$
where *r* is the exponential growth rate at very low $p_{R}\left(P,A\right)\left(t\right)$ far from the carrying capacity *C*, which is an asymptotically stable fixed point. However, the assumption that the evolution of resistance occurs independently of antibiotic consumption is difficult to justify. Although the authors of [11] have investigated correlations between *r* and *C* with consumption, we believe it makes more sense to incorporate consumption dependency directly into the dynamic model as a driving term:(4)$$\frac{dp_{R}\left(P,A\right)\left(t\right)}{dt}=r\cdot p_{R}\left(P,A\right)\left(t\right)\cdot \left(C-p_{R}\left(P,A\right)\left(t\right)\right)+\alpha \cdot \left(c_{A}\left(t\right)-c_{A}\left(t-1\right)\right).$$
Thereby, $c_{A}\left(t\right)$ is the monthly aggregated consumption density of antibiotic A∈A at month *t* with t∈{0,…,87}, using definition $c_{A}\left(0\right)-c_{A}\left(0-1\right):=0$, and parameter α is a transfer rate that reflects the strength of the impact of consumption on the growth rate of resistance. A detailed definition of $c_{A}$ is given in the following section. Hence, the growth dynamics contains four parameters, $p_{R}\left(P,A\right)\left(t=0\right),r,C,\alpha$, to be estimated per pathogen–antibiotic pair, with $p_{R}\left(P,A\right)\left(t=0\right)$ being the initial value.

The four free parameters are estimated using a maximum likelihood routine [33]. Log-normally distributed $p_{R}\left(P,A\right)$ are assumed such that the procedure reduces to least squares using logarithmized values of observations and predictions, whereby the latter result from numerical solutions of Equation (Equation 4) using the ODE solver introduced in [34].

#### 4.3.4. Consumption Density Time Series of Antibiotics in the Presence of Resistance

The dataset contains the daily doses of antibiotics administered per patient, expressed in a specific unit. The normalization of these daily doses by dividing them by the median consumption value taken over the entire observation period and across all patients results in a standardized value within the clinic, which is analogous to the general calculation of daily defined doses (DDD) but arguably closer to the often preferred recommended daily doses (RDD). Different antibiotics are therefore comparable within the clinic in terms of daily consumption, and since no interclinical comparisons are made, the type of standardization has no effect. Let $c_{A}\left(t\right)$ be the total consumption of antibiotic *A* on month *t* in standardized units with A∈A, then(5)$$p_{A}\left(t\right)=\frac{c_{A}\left(t\right)}{\sum_{B\in A}\left\{c_{B}\left(t\right)\right\}}$$
is the proportion of the consumption of antibiotic *A* in the total consumption of all antibiotics. Of note, the extent of the set of antibiotics, A, varies depending on context. Specifically, A may contain all antibiotics used in the ward at any time within the observation time, but may also be restricted to the most frequently used antibiotics to ensure a robust computation of derived functionals, such as, e.g., an entropy.

#### 4.3.5. Entropies and Diversity Measures

The concept of entropy, originally developed in physics and intimately related to the second law of thermodynamics, was applied in other disciplines over the course of the 20th century and attracted increasing attention. The prevailing interpretation of entropy in physics was that of a measure of the distance of a thermodynamic system with many microscopic degrees of freedom from the equilibrium state. The adoption in other disciplines initially started in computer science, with such an impact that entropy was retroactively understood as an observer-theoretical quantity in physics as well.

This section is dedicated to a brief introduction of the entropy and heterogeneity measures relevant for our evaluations. The individual measures are sensitive to the quantification of different aspects of heterogeneity, and a comparative presentation, therefore, appears useful. Particular attention should be paid to the temporal change in heterogeneity in the allocation of antibiotics, because according to our hypothesis, it is precisely the temporal variation that is relevant for a reduction in the incidence of resistant germs.

Within the family of entropy functionals, the Shannon entropy, named after a computer science pioneer, is probably the most common. The following definitions and explanations relate directly to the context of ABS. Let $p_{A}\left(t\right)$ be the proportion of the consumption of antibiotic *A* at time point *t* in the total consumption, then the Shannon entropy is defined as(6)$$H_{S}\left(t\right)=-\frac{1}{n}\sum_{A\in A}p_{A}\left(t\right)\ln\left(p_{A}\left(t\right)\right),$$
where *n* is the number of antibiotics ever in use and, obviously, $\sum_{A\in A}\left\{p_{A}\left(t\right)\right\}=1\;\forall t$. A simple interpretation of the Shannon entropy is that of a (inverse) distance measure, which indicates how close a certain index distribution is to a uniform distribution, i.e., the reference distribution. The entropy has its maximum at $H_{S}=1$ when the index distribution is equal to the uniform (i.e., the reference) distribution. The minimum $H_{S}=0$ is achieved for $p_{A}=1;p_{B}=0\to \forall B\neq A;A,B\in A$, i.e., if the total consumption is reduced to the consumption of a single substance *A*. In terms of information theory, $-\ln(p_{i})$ is conceived as bits of information needed to distinguish realization *i* from $1/p_{i}$ equally likely alternative realizations, hence the Shannon information corresponds to the expected information contained in the system.

Shannon entropy as defined in Equation (Equation 6) is equivalent to(7)$$KL_{uniform}\left(t\right)=1-\frac{1}{\ln(n)}\sum_{A\in A}p_{A}\left(t\right)\left[\ln\left(p_{A}\left(t\right)\right)-\ln\left(1/n\right)\right].$$

Equation (Equation 7) is a special case of the so-called Kullback–Leibler divergence defined as(8)$$KL\left(t\right)=1-\sum_{A\in A}p_{A}\left(t\right)\left[\ln\left(p_{A}\left(t\right)\right)-\ln\left({\hat{p}}_{A}\right)\right].$$
where ${\hat{p}}_{A}$ is an arbitrary reference distribution that replaces the special case of a uniform distribution ${\hat{p}}_{A}=1/n\to \forall A$. Of note, KL in Equation (Equation 8) has its maximum at KL=1; however, the given generality of ${\hat{p}}_{A}$ contradicts a definitive standardization, hence KL may extend into the negative.

In the given context, the entropy represents a heterogeneity measure for antibiotic consumption, and in fact, the Shannon entropy is structurally similar to the so-called Antibiotic Heterogeneity Index (AHI), which is defined for convenience as follows:(9)$$AHI\left(t\right)=1-\frac{n}{2(n-1)}\sum_{A\in A}\left|p_{A}\left(t\right)-{\hat{p}}_{A}\right|.$$
The reference distribution is usually chosen to be ${\hat{p}}_{A}=\frac{1}{n},\forall A\in A$, i.e., the uniform distribution. As with $H_{S}$, the minimum entropy for AHI is 0 and is achieved by reducing consumption to a single antibiotic $p_{A}=1;p_{B}=0\to \forall B\neq A;A,B\in A$. Of note, AHI−1 and 1−AHI are in use as heterogeneity indexes with the correspondingly adapted interpretations of the value ranges.

A simple consideration shows that AHI and many other heterogeneity measures do not quantify heterogeneity particularly well. Given three different antibiotics, one of which is used in a given time window with a proportion of p1=2/3. Regarding the uniform reference distribution, the same AHI is obtained for the two situations where p2=1/3, p3=0 and p2=1/6, p3=1/6, respectively. Intuitively, however, one would consider the second situation to be more homogeneous.

The reference distribution can be selected for both KL and AHI according to suitability. If $t_{0}$ is the time of the first observation, then(10)$$KL_{t_{0}}\left(t\right)=1-\sum_{A\in A}p_{A}\left(t\right)\left(\ln\left(p_{A}\left(t\right)\right)-\ln\left(p_{A}\left(t_{0}\right)\right)\right).$$
provides the temporal course of the deviations of the instantaneous shares of consumption from the initial consumption distribution. In the same way,(11)$$KL\Delta \left(t\right)\equiv H_{S}\Delta \left(t\right)=1-\sum_{A\in A}p_{A}\left(t\right)\left(\ln\left(p_{A}\left(t\right)\right)-\ln\left(p_{A}\left(t-1\right)\right)\right).$$
provides the course of the deviations of the consumption shares of a given point in time, *t*, from the consumption shares of the respective preceding point in time, t−1, assuming the existence of discrete equidistant time steps that are mapped to integer values. It is then justified to call such an entropy that quantifies subsequent changes a differential representation of heterogeneity. Likewise, a differential measure can be defined for AHI according to(12)$$AHI\Delta \left(t\right)=1-\frac{n}{2(n-1)}\sum_{A\in A}\left|p_{A}\left(t\right)-p_{A}\left(t-1\right)\right|.$$

The specific choice of the reference distribution, ${\hat{p}}_{i},\forall i\in A$, in the work of Morisawa et al. [19] is worth mentioning. In their study, they estimated AHI according to Equation (Equation 9), considering four classes of antibiotics, with a proportion of 10% being set for one of the classes, namely fluoroquinolones, while the remaining three classes each have a proportion of 30%. This setting allows clinical guidelines to be taken into account, such as the requirement that the administration of certain antibiotics be subject to restrictions.

To complete the list of the most important heterogeneity measures, the Gini Index, which is frequently used in economic analyses, should also be mentioned:(13)$$GI\left(t\right)=\frac{1}{2(n-1)}\sum_{A,B\in A,A\neq B}|p_{A}\left(t\right)-p_{B}\left(t\right)|.$$
The Gini Index scales between 0 and 1, specifically, GI=0 if $p_{A}=1/n,\forall A\in A$.

In ecology, exponentials of the entropies are the basis for the definition of diversities, thus merely representing monotonous transformations of heterogeneities. In the given context, diversity may be a more intuitively understandable naming of a property regarding the consumption of antibiotics, as diversity is directly associated with a “healthy”, i.e., more robust and resilient system state with respect to antimicrobial resistance. At least this corresponds to the hypothesis to be examined here. For a comprehensive discussion of diversity and heterogeneity measures in the context of ABS, please refer to our introductory methodological paper [12].

In the same way as for Equations (Equation 1) and (Equation 2), all entropy time series introduced above, as they are ultimately fed into the evaluations performed, are smoothed using a moving average function (central, partially at both ends) with a window of 24 months in order to curb the peaks.

#### 4.3.6. Phase Space Reconstruction

If we consider the time-dependent courses of entropy (e.g., AHI(t)) and share of resistance ($p_{R}\left(t\right)$) as coupled variables of a dynamic system, then the manifold $\left\{AHI\left(t\right),p_{R}\left(t\right)\right\}$ provides a trajectory in the phase space spanned by the two variables. This makes it easy to recognize and analyze phase space structures. Furthermore, fragmented regressions of the individual variables on time, i.e., the estimation of change points in time, can be used to illustrate multiphase time courses very clearly. We use the R package (version 1.0.25.1) *chngpt* [35] for this purpose. Such multiphase processes can result, for example, from relatively rapidly implemented antibiotic administration strategies.

## 5. Conclusions

The evolution of the proportion of pathogen isolates that prove resistant to a specific antibiotic challenge does not follow simple saturation dynamics, nor does it follow dynamics that are univariably dependent on the consumption of the specific antibiotic, with the exception of a few individual cases of pathogen–antibiotic pairs, particularly *P. aeruginosa*-meropenem. However, modeling using higher-dimensional, coupled dynamics is not feasible given the available data. A promising modeling approach that can be operationalized in terms of antibiotic stewardship is the description of the aggregate incidence of resistance as a function of a differential heterogeneity functional. In addition, this is precisely where our key message comes in: the decisive driving force behind the reduction in resistance emergence is not the current heterogeneity, but the extent of its constant change. However, strictly planned cycling is not recommended; rather, the heuristic tacit knowledge-driven clinical cycling that takes place anyway should be complemented by a targeted operational approach, in other words, the integration of clinical expertise with model-based prediction.

## Figures and Tables

**Figure 1 antibiotics-14-01266-f001:**
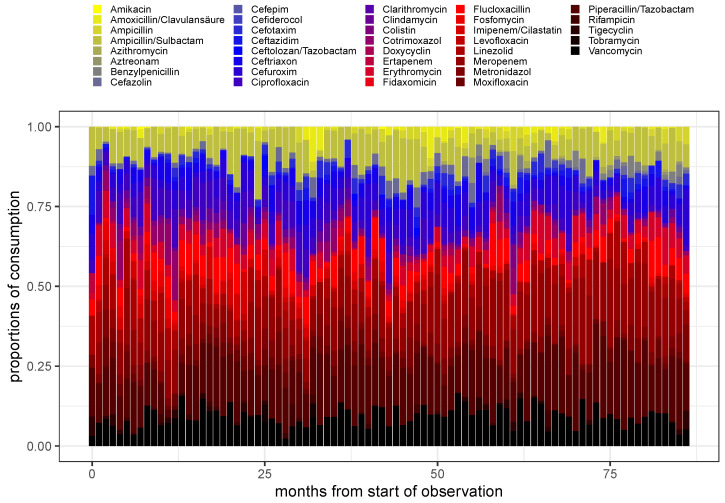
Stacked barplot of the proportions of consumption for all (not too rarely) administered antimicrobials over time using monthly aggregated consumptions.

**Figure 2 antibiotics-14-01266-f002:**
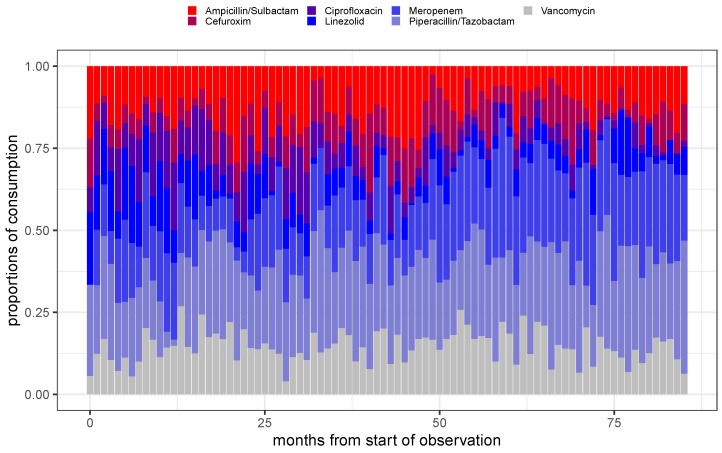
Stacked barplot of the proportions of consumption for the seven most frequent administered antimicrobials over time using monthly aggregated consumptions.

**Figure 3 antibiotics-14-01266-f003:**
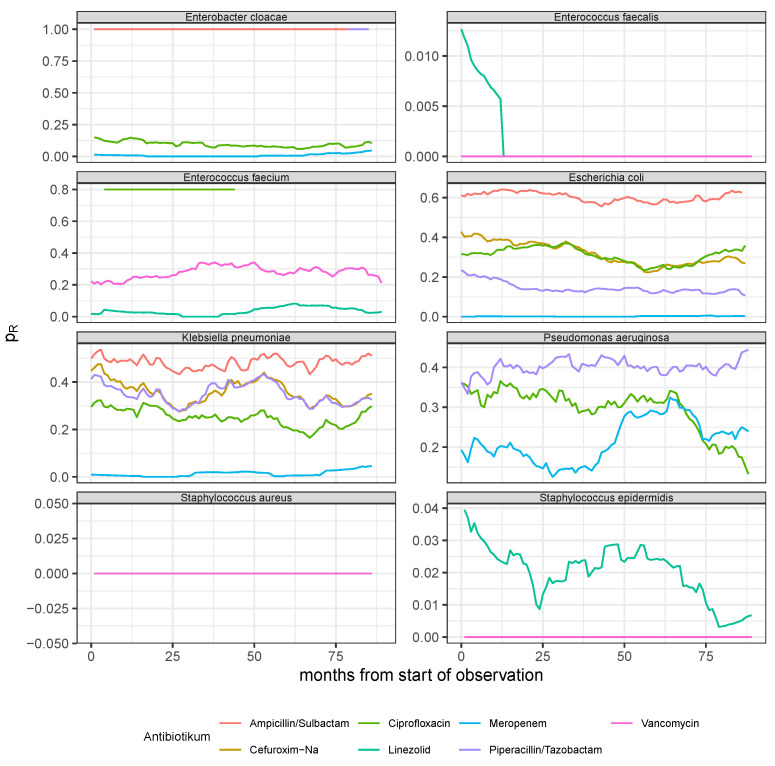
Observed resistance rate ($p_{R}$, see Equation (Equation 1)) time series derived from monthly aggregated antibiogram results of the eight most common pathogens (see panel headers) with respect to the experimentally applied substances out of the seven most frequently administered antibiotics (cf. legend for the color code).

**Figure 4 antibiotics-14-01266-f004:**
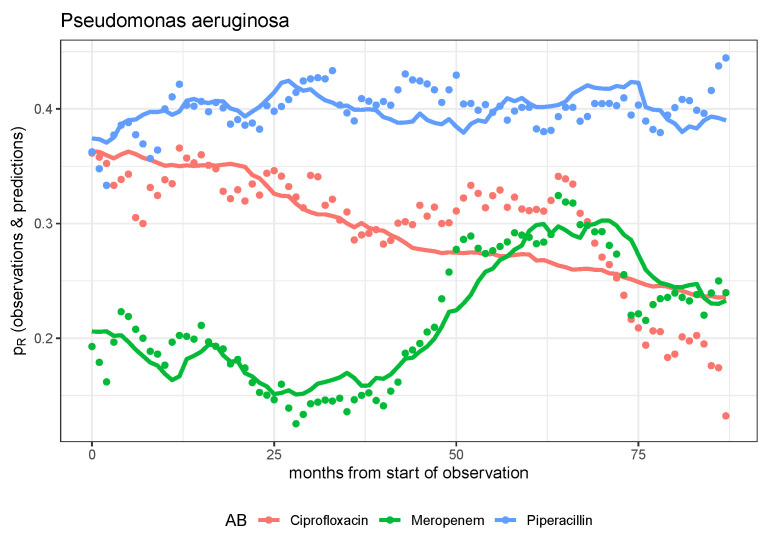
Observed time series and model predictions of the three rates of resistance of *P. aeruginosa* to ciprofloxacin, meropenem, and piperacillin, respectively.

**Figure 5 antibiotics-14-01266-f005:**
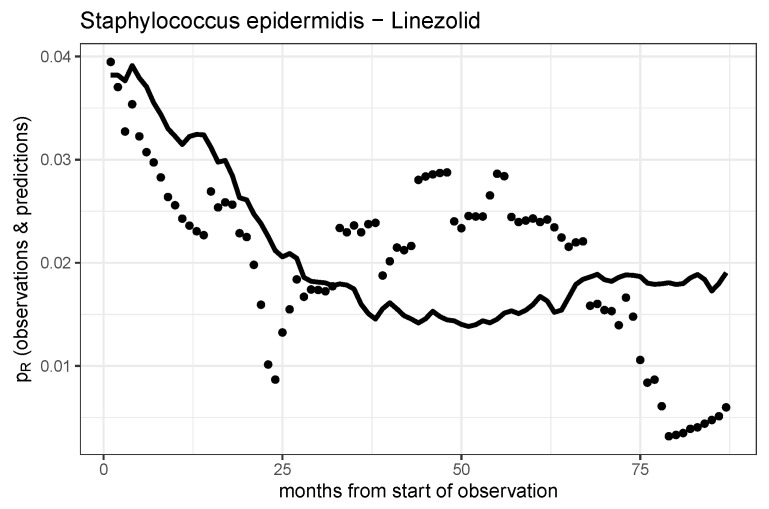
Observed time series and model prediction of the rate of resistance of *S. epidermidis* to linezolid.

**Figure 6 antibiotics-14-01266-f006:**
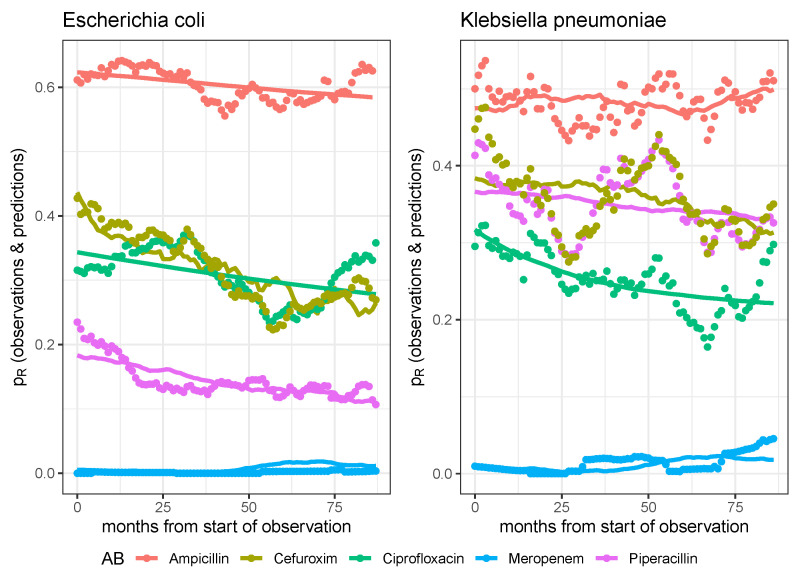
Observed time series and model predictions of the respective five rates of resistance of *E. coli* (**left panel**) and *K. pneumoniae* (**right panel**) to ciprofloxacin, meropenem, piperacillin, cefuroxim, and ampicillin, respectively.

**Figure 7 antibiotics-14-01266-f007:**
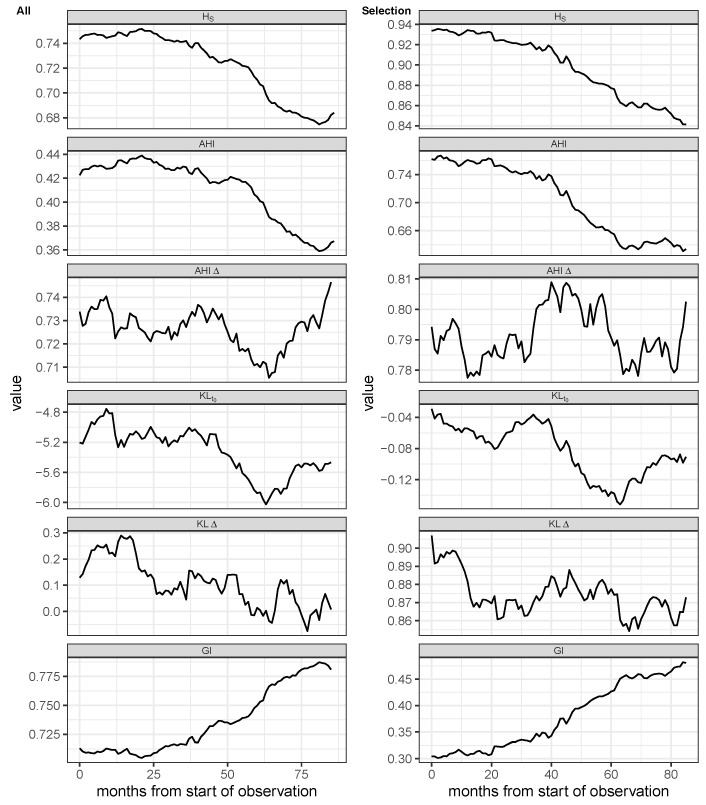
Shannon entropy ($H_{S}$), Antibiotic Heterogeneity Index (AHI) with uniform distribution as reference, AHI with distribution at previous month as reference (AHIΔ), two versions of Kullback–Leibler divergence, one with distribution at the first observation as reference (${\mathrm{KL}}_{t_{0}}$), and one with the distribution at previous month as reference (KLΔ), and Gini Index (GI) of monthly aggregated proportional consumption time series. First column: All antimicrobials excluding antivirals, antifungals, and rarely administered substances. Second column: Subset containing the seven most frequent antibiotics. The entropy time series is based on monthly aggregated antibiotic consumption and was smoothed via a moving average over 12 months.

**Figure 8 antibiotics-14-01266-f008:**
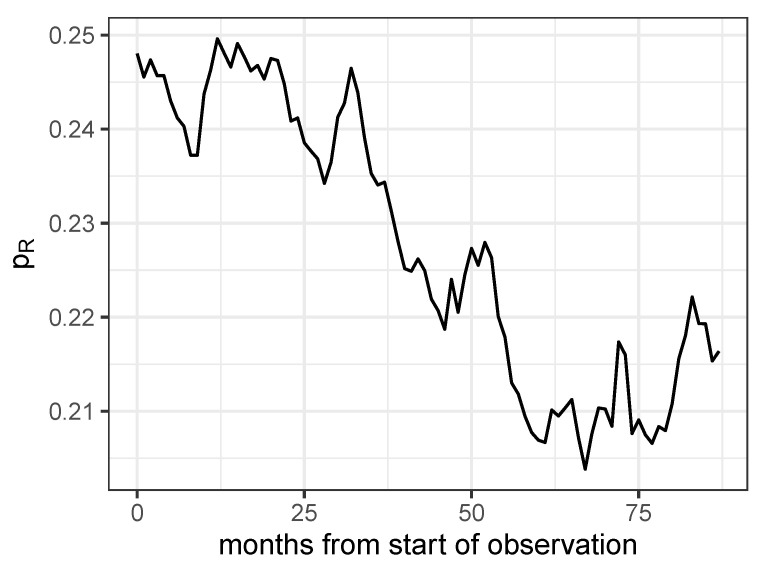
Proportion of antibiograms detected as resistant for all combinations of the eight most frequently investigated pathogens and the seven most frequently administered antibiotics.

**Figure 9 antibiotics-14-01266-f009:**
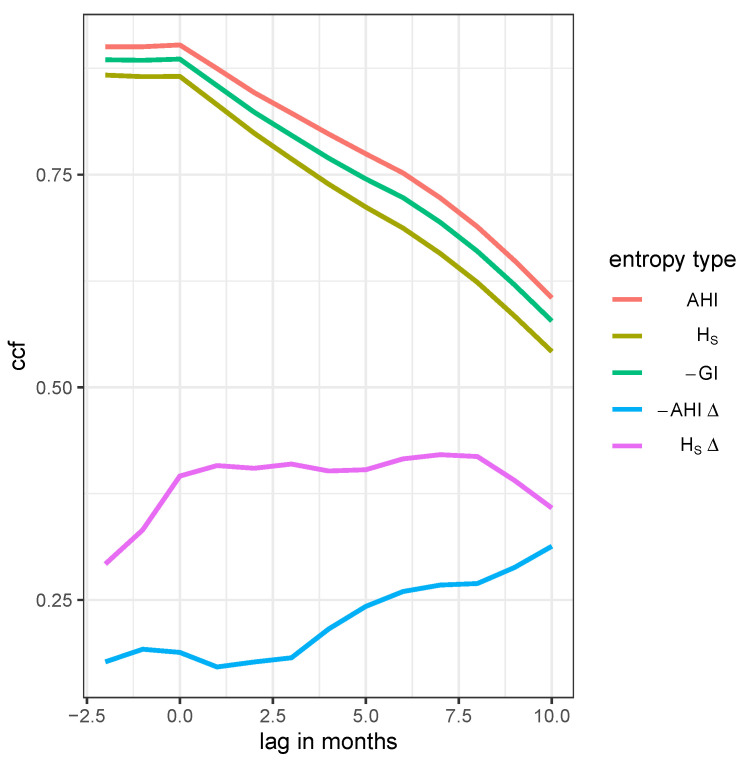
Time lag cross-correlation function (CCF) between the proportion of resistance and one of five entropies: AHI, Shannon entropy ($H_{S}$), Gini Index (GI), differential AHIΔ, and the differential Shannon entropy $H_{S}\Delta$, respectively, as annotated in the legend labels.

**Figure 10 antibiotics-14-01266-f010:**
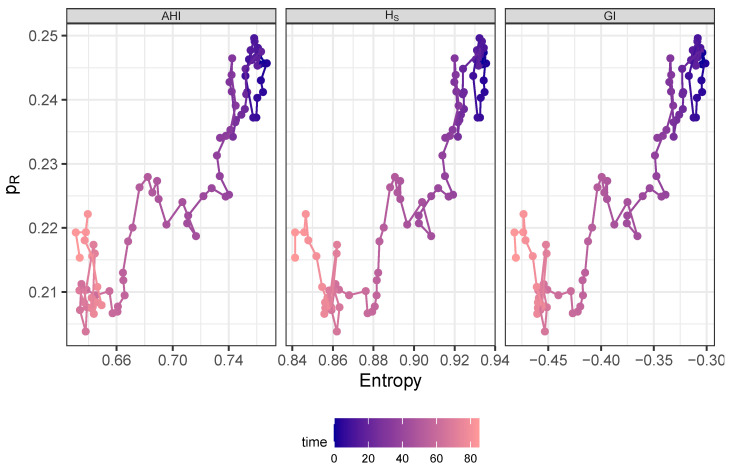
Trajectory in phase space spanned by AHI (**left panel**), Shannon entropy (**middle panel**), or negative Gini Index (**right panel**), respectively, and share of resistance drawn with color gradient by time starting at dark blue and ending at bright red.

**Table 1 antibiotics-14-01266-t001:** The 9 most frequently administered antimicrobials.

Antibiotic	*n* Days	*n* Individuals
piperacillin/tazobactam i.v.	8227	1752
meropenem i.v.	6643	1011
caspofungin i.v.	3598	439
vancomycin i.v.	3455	637
ampicillin/sulbactam i.v.	2927	903
flucloxacillin i.v.	2448	406
linezolid i.v.	2434	371
ciprofloxacin i.v.	2384	547
cefuroxim i.v.	2265	596

**Table 2 antibiotics-14-01266-t002:** Summary table of demographic characteristics stratified by antibiotic treatment flag. Subjects are ward stays.

Characteristic	Overall ^1^ N=9201	w/o AB ^1^ N=4281	with AB ^1^ N=4920	*p*-Value ^2^
**Sex**				<0.001
female	4141 (45%)	2120 (50%)	2021 (41%)	
male	5057 (55%)	2161 (50%)	2896 (59%)	
Unknown	3	0	3	
**Age**	66 (54, 78)	66 (52, 78)	66 (55, 77)	0.4
**Age Group**				0.5
<median	4436 (48%)	2081 (49%)	2355 (48%)	
≥median	4765 (52%)	2200 (51%)	2565 (52%)	
**Destination**				<0.001
death	1545 (17%)	487 (11%)	1058 (22%)	
external hospital	3121 (34%)	1405 (33%)	1716 (35%)	
home	4535 (49%)	2389 (56%)	2146 (44%)	
**Hospital Duration**	14 (7, 26)	9 (4, 16)	21 (11, 37)	<0.001
**ICU Duration**	2 (1, 5)	1 (1, 2)	4 (1, 9)	<0.001

^1^ *n* (%); Median (Q1, Q3). ^2^ Pearson’s Chi-squared test; Wilcoxon rank sum test.

**Table 3 antibiotics-14-01266-t003:** Frequency table showing the number of occurrences of each sensitivity class S, I, R, and n.a., respectively.

Sensitivity	Challenges	Distinct IDs
n.a.	537	148
I	12,309	2477
R	41,525	2951
S	101,223	3406

**Table 4 antibiotics-14-01266-t004:** Ten most frequently isolated pathogens.

Pathogen	n
*Escherichia coli*	3091
*Staphylococcus epidermidis*	2664
*Candida albicans*	2299
*Enterococcus faecium*	1745
*Staphylococcus aureus*	1579
*Klebsiella pneumoniae*	1336
*Candida glabrata*	1265
*Pseudomonas aeruginosa*	1126
*Enterococcus faecalis*	644
*Enterobacter cloacae*	614

**Table 5 antibiotics-14-01266-t005:** Maximum likelihood parameter estimates of the dynamic model predicting the resistance rate time courses of pathogen–antibiotic pairs. Abbreviations used: AB = antibiotic, patho = pathogen, Cipro = ciprofloxacin, Mero = meropenem, Piper = piperacillin/tazobactam, Line = Linezolid, P.aeru = *Pseudomonas aeruginosa*, S.epiderm = *Staphylococcus epidermidis*. The logarithmic parameters were actually estimated with the corresponding estimated values “log(estimates)”. Therefore, the standard errors stderr refer to these values. NaN refers to a divergent estimate.

Patho-AB-Pair	Parameter	Estimate	log(estimate)	Stderr	*p*-Value
P.aeru-Cipro	$f_{0}$	0.363	−1.013	0.470	0.031
	*r*	0.008	−4.801	NaN	NaN
	*C*	0.163	−1.815	1.479	0.220
	α	0.003	−5.883	1.668	0.000
P.aeru-Mero	$f_{0}$	0.206	−1.581	0.633	0.013
	*r*	0.014	−4.270	9.670	0.659
	*C*	0.122	−2.107	3.666	0.565
	α	0.004	−5.567	1.930	0.004
P.aeru-Piper	$f_{0}$	0.380	−0.967	0.344	0.005
	*r*	0.001	−7.020	8.582	0.413
	*C*	0.182	−1.703	2.221	0.443
	α	0.002	−6.388	4.721	0.176
S.epiderm-Line	$f_{0}$	0.038	−3.265	4.104	0.426
	*r*	0.009	−4.665	24.934	0.852
	*C*	0.021	−3.885	20.238	0.848
	α	0.001	−7.314	7.014	0.297

**Table 6 antibiotics-14-01266-t006:** Maximum likelihood parameter estimates of the dynamic model predicting the resistance rate time courses of pathogen–antibiotic pairs. Abbreviations used: AB = antibiotic, patho = pathogen, Cipro = ciprofloxacin, Mero = meropenem, Piper = piperacillin/tazobactam, Ampi = ampicillin/sulbactam, Cefu = cefuroxim, K.pneu = *Klebsiella pneumoniae*, E.coli = *Escherichia coli*. The logarithmic parameters were actually estimated with the corresponding estimated values “log(estimates)”. Therefore, the standard errors stderr refer to these values. NaN refers to a divergent estimate.

Patho-AB-Pair	Parameter	Estimate	log(estimate)	Stderr	*p*-Value
E.coli-Cipro	$f_{0}$	0.344	−1.068	0.646	0.098
	*r*	0.008	−4.774	5.802	0.411
	*C*	0.028	−3.572	13.496	0.791
	α	0.000	−10.116	18.798	0.590
E.coli-Mero	$f_{0}$	0.005	−5.267	NaN	NaN
	*r*	0.001	−6.779	8.409	0.42
	*C*	0.288	−1.243	NaN	NaN
	α	0.000	−7.745	NaN	NaN
E.coli-Piper	$f_{0}$	0.183	−1.697	1.245	0.173
	*r*	0.056	−2.884	3.829	0.451
	*C*	0.010	−4.656	8.209	0.571
	α	0.001	−6.965	8.408	0.407
E.coli-Ampi	$f_{0}$	0.623	−0.473	0.317	0.136
	*r*	0.001	−6.520	NaN	NaN
	*C*	0.095	−2.357	NaN	NaN
	α	0.000	−9.560	28.409	0.736
E.coli-Cefu	$f_{0}$	0.436	−0.829	1.074	0.440
	*r*	0.165	−1.801	6.518	0.782
	*C*	0.270	−1.310	1.349	0.331
	α	0.008	−4.861	11.319	0.668
K.pneu-Cipro	$f_{0}$	0.317	−1.149	0.848	0.175
	*r*	0.097	−2.333	5.190	0.653
	*C*	0.212	−1.552	1.647	0.346
	α	0.000	−8.423	20.356	0.679
K.pneu-Mero	$f_{0}$	0.008	−4.854	NaN	NaN
	*r*	0.012	−4.438	NaN	NaN
	*C*	0.186	−1.680	NaN	NaN
	α	0.000	−7.708	2.709	0.004
K.pneu-Piper	$f_{0}$	0.366	−1.005	0.601	0.095
	*r*	0.007	−5.030	7.394	0.496
	*C*	0.075	−2.595	NaN	NaN
	α	0.001	−7.384	4.123	0.073
K.pneu-Ampi	$f_{0}$	0.475	−0.745	0.298	0.012
	*r*	0.004	−5.469	6.965	0.432
	*C*	0.649	−0.432	0.539	0.423
	α	0.002	−6.243	6.976	0.371
K.pneu-Cefu	$f_{0}$	0.384	−0.958	0.691	0.166
	*r*	0.015	−4.190	7.140	0.557
	*C*	0.141	−1.959	3.101	0.527
	α	0.004	−5.654	7.407	0.445

## Data Availability

Data available on request due to legal and ethical restrictions.

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
