# Peer review of "Association Between Differential Heterogeneity of Antibiotics Consumption and Share of Resistant Pathogens and Its Implication for Antibiotic Stewardship in a German Hospital Intensive Care Unit"

_antibiotics, 2025, doi:10.3390/antibiotics14121266_

Round 1

Reviewer 1 Report

Comments and Suggestions for Authors

This article summarizes authors’ analyses of secondary data from the surgical intensive care unit of a University Hospital in Bochum, recorded over a 7 year time period, in order to reveal the correlation between antibiotic use and the development of pathogen resistance to these antibiotics. This is a re-evaluation of the data recently published by the same authors in a more comprehensive and dynamic approach. The shortcomings of linear and uncoupled approaches, as well as inapplicability of the differential equations based models using the available data are well explained. Evaluating their data with a consideration of heterogeneity of antibiotic consumption, the authors concluded that the integration of clinical expertise with model-based prediction is necessary. The topic is very important and timely and will therefore attract attention of readers from various fields. Thus, the publication of this very important article is highly recommended. There are only several minor points that need to be addressed before the manuscript can be accepted for publication.

  1. Lines 190-191: The mention of an article in a sentence this way sounds interesting. I would recommend rephrasing as: …. published by Gerding et al. [15]….

Same holds for Lines 217-218: …..  study by Sandiumenge et al. [17]…

  1. Table 2: Number in brackets must be explained in the footnote for Age, Hospital Duration and ICU Duration data.
  2. Table 4: Pathogen names must be italicized.
  3. While the color transitions in Figures 1 and 2 are aesthetically pleasing, they make the data presented in these tables difficult to grasp. Using different shades of color side by side, rather than using consecutive colors in the color scale for the corresponding agents in the figures, would make the data easier to understand.
  4. Figure 3: the y-axis must be described.

Author Response

This article summarizes authors’ analyses of secondary data from the surgical intensive care unit of a University Hospital in Bochum, recorded over a 7 year time period, in order to reveal the correlation between antibiotic use and the development of pathogen resistance to these antibiotics. This is a re-evaluation of the data recently published by the same authors in a more comprehensive and dynamic approach. The shortcomings of linear and uncoupled approaches, as well as inapplicability of the differential equations based models using the available data are well explained. Evaluating their data with a consideration of heterogeneity of antibiotic consumption, the authors concluded that the integration of clinical expertise with model-based prediction is necessary. The topic is very important and timely and will therefore attract attention of readers from various fields. Thus, the publication of this very important article is highly recommended. There are only several minor points that need to be addressed before the manuscript can be accepted for publication.

  1. Lines 190-191: The mention of an article in a sentence this way sounds interesting. I would recommend rephrasing as: …. published by Gerding et al. [15]….

Same holds for Lines 217-218: …..  study by Sandiumenge et al. [17]…

Author Response 1:

Thanks for pointing us to this odd kind of citation. For the first preparation round of the manuscript we used a different LaTeX-citation style in the form of “First Author et al., Year” instead of the enumeration in brackets. We corrected the corresponding passages accordingly.

  1. Table 2: Number in brackets must be explained in the footnote for Age, Hospital Duration and ICU Duration data.

Author Response 2:

Due to a mistake in LaTeX typesetting, the footnotes have not been printed. We corrected this mistake.

  1. Table 4: Pathogen names must be italicized.

Author Response 3:

This oversight has been corrected.

  1. While the color transitions in Figures 1 and 2 are aesthetically pleasing, they make the data presented in these tables difficult to grasp. Using different shades of color side by side, rather than using consecutive colors in the color scale for the corresponding agents in the figures, would make the data easier to understand.

Author Response 4:

We agree with the Reviewer that the color palette was very poorly chosen. However, it became apparent that placing the bars for the respective substances side by side for each month resulted in a completely illegible blur. We therefore decided to achieve visual separation by choosing a better color palette.

  1. Figure 3: the y-axis must be described.

Author Response 5:

Thanks again for pointing us to this accidental omission. We accordingly added an explanation in the figure legend.

General Response:

We would like to thank the reviewer for their efforts, their generally positive assessment of our manuscript, and their valuable and justified corrections.

Reviewer 2 Report

Comments and Suggestions for Authors

The manuscript presents commendable insights into how differential antibiotic consumption patterns may influence resistance dynamics in an intensive care setting. However, it is lengthy, complex, and at times lacks clarity in its methodological exposition and interpretation. I recommend the following revisions before it can be considered for publication:

  1. Several key studies central to antibiotic heterogeneity, cycling versus mixing strategies, and consumption–resistance dynamics are missing and should be incorporated to properly contextualize the manuscript’s claims. In particular, the large multicentre cluster-randomized trial by van Duijn et al. (DOI: 10.1016/S1473-3099(18)30056-2) evaluating cycling vs. mixing in ICUs, along with subsequent systematic reviews, represents critical clinical evidence that must be acknowledged when discussing policy or causal implications. The manuscript would also benefit from referencing foundational theoretical work such as Bergstrom et al. (https://doi.org/10.1073/pnas.0402298101) on evolutionary arguments for mixing, and methodological precedents including threshold/logistic time-series models used in antimicrobial resistance surveillance. Incorporating these studies will strengthen the justification for the authors’ approach, clarify the manuscript’s novelty, and ensure that broader conclusions are appropriately supported.
  2. I did not find a clear rationale for selecting a logistic growth–based resistance model. While the manuscript presents the model and its fits, it does not explain why a logistic form was chosen over other established approaches in AMR modeling (e.g., mixed-effects time-series models, state-space formulations, or agent-based simulations). Because several interpretations in the Results and Discussion rely on how well or poorly the logistic model fits specific bug–drug trajectories, the authors should briefly justify this modeling choice and clarify the biological and statistical assumptions inherent in the logistic formulation.
  3. The manuscript would benefit from more rigorous validation of the logistic model. The fits shown for certain bug–drug pairs are described as strong, but the assessment is qualitative, as no quantitative goodness-of-fit metrics (e.g., R², RMSE, AIC, or cross-validation) are provided. Without such metrics or sensitivity analyses testing the robustness of parameter estimates, the evaluation of model performance remains anecdotal. Including standard fit statistics and at least one robustness check would substantially strengthen the modeling section.
  4. The authors infer that the observed delayed correlations (5–7 months) between entropy changes and resistance may indicate functional dependence. However, correlation alone cannot establish causality. A discussion of potential confounders, such as infection control interventions, ICU case mix, diagnostic intensity, or underlying pathogen ecology that may influence both antibiotic consumption and resistance trends, would improve the rigor and balance of the interpretation.
  5. Abstract is overly long

Author Response

The manuscript presents commendable insights into how differential antibiotic consumption patterns may influence resistance dynamics in an intensive care setting. However, it is lengthy, complex, and at times lacks clarity in its methodological exposition and interpretation. I recommend the following revisions before it can be considered for publication:

  1. Several key studies central to antibiotic heterogeneity, cycling versus mixing strategies, and consumption–resistance dynamics are missing and should be incorporated to properly contextualize the manuscript’s claims. In particular, the large multicentre cluster-randomized trial by van Duijn et al. (DOI: 10.1016/S1473-3099(18)30056-2) evaluating cycling vs. mixing in ICUs, along with subsequent systematic reviews, represents critical clinical evidence that must be acknowledged when discussing policy or causal implications. The manuscript would also benefit from referencing foundational theoretical work such as Bergstrom et al. (https://doi.org/10.1073/pnas.0402298101) on evolutionary arguments for mixing, and methodological precedents including threshold/logistic time-series models used in antimicrobial resistance surveillance. Incorporating these studies will strengthen the justification for the authors’ approach, clarify the manuscript’s novelty, and ensure that broader conclusions are appropriately supported.

Author Response 1:

The discussed and contextualized cycling and mixing strategies within the Introduction. We also did refer to the RCT by Duijin et al., however, we unfortunately missed to properly cite their work. We thank the Reviewer pointing to this accidental omission. Special thanks are due to the reviewer for referring to Bergstrom et al. Indeed, this reference highlights one of the important points we wish to make, namely that a significant difference in effect can be expected between informed and rational clinical cycling and scheduled cycling. We have included a detailed explanation of this at the end of section 2.3, where we also properly cited the works by Duijin et al. and Bergstrom et al.

  1. I did not find a clear rationale for selecting a logistic growth–based resistance model. While the manuscript presents the model and its fits, it does not explain why a logistic form was chosen over other established approaches in AMR modeling (e.g., mixed-effects time-series models, state-space formulations, or agent-based simulations). Because several interpretations in the Results and Discussion rely on how well or poorly the logistic model fits specific bug–drug trajectories, the authors should briefly justify this modeling choice and clarify the biological and statistical assumptions inherent in the logistic formulation.

Author Response 2:

As described in the manuscript, we chose the logistic growth model as the basic model for resistance development in order to explicitly refer to the brand-new publication by Emons et al. 2025, who applied the model extensively to resistance development in various countries. We wanted to compare our results with their modeling results. As a basic model, the logistic model with two parameters is even more flexible than, for example, a linear or even constant model with only one parameter. When modeling the specific temporal progression of bug-drug pairs, the rates of consumption changes dominated in terms of significance, although the pairwise modeling was not particularly well adapted overall. This was to be demonstrated.

  1. The manuscript would benefit from more rigorous validation of the logistic model. The fits shown for certain bug–drug pairs are described as strong, but the assessment is qualitative, as no quantitative goodness-of-fit metrics (e.g., R², RMSE, AIC, or cross-validation) are provided. Without such metrics or sensitivity analyses testing the robustness of parameter estimates, the evaluation of model performance remains anecdotal. Including standard fit statistics and at least one robustness check would substantially strengthen the modeling section.

Author Response 3:

In our opinion, a goodness-of-fit metric in the context of fitting a dynamic model to an observed time series is only meaningful when comparing two (nested) model variants. In the case of fitting one n-parameter model it is essential, how significant each of the n parameter contributes to the fit. In principle, a comparison of each fitted model to a constant function given by the time average would be possible, however we doubt that any relevant information could thereby be drawn beyond the qualitative overall assessment plus the impact of the parameters.

  1. The authors infer that the observed delayed correlations (5–7 months) between entropy changes and resistance may indicate functional dependence. However, correlation alone cannot establish causality. A discussion of potential confounders, such as infection control interventions, ICU case mix, diagnostic intensity, or underlying pathogen ecology that may influence both antibiotic consumption and resistance trends, would improve the rigor and balance of the interpretation.

Author Response 4:

We thank the reviewer for reminding us of the causality problem. We are fully aware of this problem. Functional dependence definitely does not refer to explicitly given causality; rather, we are referring to Ernst Mach's view that there are neither real causes nor causal relationships in nature, but only functional relationships. In this context, causality is given as an apriori. Not even the simple models that describe bug-drug pairs are really “mono-causal”, as we incorrectly wrote, thus we changed that phrase to “univariable”. On a more holistic structural level of entropies, however, as our explanation points out, the delay correlation provides a temporal direction of effect that can be used for operational measures. In addition, we have emphasized the concept of clinical cycling from the outset. Specifically, this means that we assume that clinical expertise and other clinical conditions influence the prescription of antibiotics and thus the development of resistance per se. The observed heterogeneity or change in heterogeneity is a surrogate that depends on or correlates with all these confounders per se. In the given context of functional relationships, the attempt to derive a strategy for rational administering antibiotics represents an approach to deriving order or control parameters within a real-world-setting. In other words, the inclusion of diagnostic intensity and underlying pathogen ecology is an intrinsic property, i.e. the very essence of our modeling approach.

  1. Abstract is overly long

Author Response 5:

We have shortened the abstract.

Round 2

Reviewer 2 Report

Comments and Suggestions for Authors

The revised manuscript is acceptable for publication